# Ambroxol-Enhanced Frequency and Amplitude of Beating Cilia Controlled by a Voltage-Gated Ca^2+^ Channel, Cav1.2, via pH_i_ Increase and [Cl^−^]_i_ Decrease in the Lung Airway Epithelial Cells of Mice

**DOI:** 10.3390/ijms242316976

**Published:** 2023-11-30

**Authors:** Takashi Nakahari, Chihiro Suzuki, Kotoku Kawaguchi, Shigekuni Hosogi, Saori Tanaka, Shinji Asano, Toshio Inui, Yoshinori Marunaka

**Affiliations:** 1Research Unit for Epithelial Physiology, Research Organization of Science and Technology, BKC, Ritsumeikan University, Kusatsu 525-8577, Japan; ymr18005@fc.ritsumei.ac.jp; 2Medical Research Institute, Kyoto Industrial Health Association, Kyoto 604-8472, Japan; 3Laboratory of Pharmacotherapy, Faculty of Pharmacy, Osaka Medical and Pharmaceutical University, Takatsuki 569-1094, Japan; heureux.petite.fille@gmail.com (C.S.); saori.tanaka@ompu.ac.jp (S.T.); 4Department of Molecular Physiology, Faculty of Pharmacy, BKC, Ritsumeikan University, Kusatsu 525-8577, Japan; k-kawagu@fc.ritsumei.ac.jp (K.K.); ashinji@fc.ritsumei.ac.jp (S.A.); 5Department of Clinical and Translational Physiology, Kyoto Pharmaceutical University, Kyoto 607-8414, Japan; hosogi@mb.kyoto-phu.ac.jp; 6Saisei Mirai Clinics, Moriguchi 570-0012, Japan; t-inui@saisei-mirai.or.jp

**Keywords:** voltage-gated Ca^2+^ channels, [Ca^2+^]_i_, intracellular pH, intracellular Cl^−^ concentration, cell volume, Ca_V_1.2, airway ciliated epithelial cells

## Abstract

Ambroxol (ABX), a frequently prescribed secretolytic agent which enhances the ciliary beat frequency (CBF) and ciliary bend angle (CBA, an index of amplitude) by 30%, activates a voltage-dependent Ca^2+^ channel (Ca_V_1.2) and a small transient Ca^2+^ release in the ciliated lung airway epithelial cells (c-LAECs) of mice. The activation of Ca_V_1.2 alone enhanced the CBF and CBA by 20%, mediated by a pH_i_ increase_i_ and a [Cl^−^]_i_ decrease in the c-LAECs. The increase in pH_i_, which was induced by the activation of the Na^+^-HCO_3_^−^ cotransporter (NBC), enhanced the CBF (by 30%) and CBA (by 15–20%), and a decrease in [Cl^−^]_i_, which was induced by the Cl^−^ release via anoctamine 1 (ANO1), enhanced the CBA (by 10–15%). While a Ca^2+^-free solution or nifedipine (an inhibitor of Ca_V_1.2) inhibited 70% of the CBF and CBA enhancement using ABX, Ca_V_1.2 enhanced most of the CBF and CBA increases using ABX. The activation of the Ca_V_1.2 existing in the cilia stimulates the NBC to increase pH_i_ and ANO1 to decrease the [Cl^−^]_i_ in the c-LAECs. In conclusion, the pH_i_ increase and the [Cl^−^]_i_ decrease enhanced the CBF and CBA in the ABX-stimulated c-LAECs.

## 1. Introduction

Mucociliary clearance (MC), which is a host defense mechanism of the lungs, consists of a surface fluid layer (surface mucous layer (SML) and periciliary layer (PCL)) and beating cilia lining the airway surface [1,2,3]. Inhaled small particles, such as bacteria, viruses and chemicals, are entrapped by the SML and swept away toward the oropharynx by the beating cilia in the PCL. The MC is compared to a belt conveyor system in sweeping away inhaled small particles from the airways, and the beating cilia are the engine that drives MC [1,3,4]. Impairment of the beating cilia, such as Kartagener syndrome or primary ciliary dyskinesia, causes serious respiratory diseases [1,3]. Therefore, the drugs activating beating cilia are of particular importance to prevent or improve respiratory diseases. Ambroxol (ABX), a frequently prescribed secretolytic agent for respiratory diseases, which increases [Ca^2+^]_i_ by activating a voltage-gated Ca^2+^ channel, Ca_V_1.2, in the ciliated lung airway epithelial cells (c-LAECs), has been shown to increase the CBF (ciliary beat frequency) and CBA (ciliary bend angle, an index of amplitude) [5,6]. However, the Ca^2+^-regulated signals following Ca_V_1.2 activation, which may enhance CBF and CBA, remain uncertain.

The beating cilia in the airways express many ion channels, including Ca^2+^-permeable channels, such as Ca_V_1.2, transient receptor potential (TRP) V4, TRPA1 and TRPM8 [6,7,8]. However, it is still controversial how these channels enhance the ciliary beating. The beating cilia in the airways are activated by many substances, such as cAMP, Ca^2+^, ATP, β_2_-agonists, Cl^−^ and H^+^ (intracellular pH (pH_i_)) [4,6,7,8,9,10,11,12]. Among them, Ca^2+^ is an important ion that activates the beating cilia [3,4,6,7,8]. ABX stimulates Ca_V_1.2 and a small transient Ca^2+^ release from the acidic stores in the c-LAECs [6,13]. These observations suggest that Ca_V_1.2 plays an important role in the activation of ciliary beating during ABX stimulation. Moreover, ABX has been shown to stimulate anion secretion in lung epithelial cell lines [14]. The activation of Ca_V_1.2 may stimulate Cl^−^ secretion in the c-LAECs, leading to a decrease in the intracellular Cl^−^ concentration ([Cl^−^]_i_), since anoctamine 1 (ANO1) exists in nasal ciliated epithelial cells [12]. Moreover, Saito et al. suggested that ABX stimulates a pH_i_ increase in c-LAECs [6]. Thus, Ca_V_1.2 activation stimulated via ABX may increase the pH_i_ and decrease the [Cl^−^]_i_ in c-LAECs. Previous studies have demonstrated that a pH_i_ increase and an [Cl^−^]_i_ decrease enhance the CBF and CBA in airway ciliary cells [9,10,12]. The activation of Ca_V_1.2 existing in the cilia may increase pH_i_ to enhance the CBF and CBA and decrease [Cl^−^]_i_ to enhance the CBA.

Cilia have two functionally distinct molecular motors, namely outer dynein arms (ODAs) and inner dynein arms (IDAs), which regulate the ciliary beat frequency (CBF) and ciliary bend angle (CBA), respectively [15,16]. Previous studies have demonstrated that a CBA increase, in addition to a CBF increase, enhances ciliary transport in the airway [12], although the CBF has been used to assess the activity of ciliary beating [3,4]. The signaling pathways regulating the CBF and CBA have been shown to be different [9,11,12]. ABX has been shown to enhance the CBF and CBA [6], suggesting that the activation of Ca_V_1.2 may trigger two signaling pathways, increasing the CBF and CBA.

In this study, we examined the effects of Ca_V_1.2 on the enhancement of the CBF and CBA in the ABX-stimulated c-LAECs of mice. In experiments using two solutions with and without CO_2_/HCO_3_^−^, we found that the Ca_V_1.2 activated using ABX stimulated two signaling pathways to increase the CBF and CBA, CO_2_/HCO_3_^−^-dependent and CO_2_/HCO_3_^−^-independent. This study is designed to clarify how Ca_V_1.2 activates the two signalling pathways increasing the CBF and CBA in c-LAECs.

## 2. Results

In unstimulated c-LAECs perfused with the HCO_3_^−^-containing control solution, the CBF and CBA were 8–10 Hz and 70°–90°, respectively [9,11]. Saito et al. demonstrated that ABX increases the CBF and CBA in a concentration-dependent manner and it maximally increases the CBF and CBA at 10 µM [6]. In this study, the concentration of ABX used was 10 µM. In this study, we used a nominally Ca^2+^-free solution. Because an EGTA-containing Ca^2+^-free solution increases the CBF by inhibiting the Ca^2+^-dependent PDE1A existing in the metabolon, regulating the ODAs (CBF) in the cilia [11]. To inhibit Ca_V_1.2, we used nifedipine [6]. To increase the pH_i_, we applied a CO_2_/HCO_3_^−^-free solution, which enhances the CBF and CBA [9,10]. The application of a CO_2_/HCO_3_^−^-free solution is a well-known procedure to increase pH_i_,.

### 2.1. ABX-Stimulated Cellular Events

#### 2.1.1. ABX-Stimulated Increases in the CBF and CBA

Appendix A show video images of a c-LAEC before and 15 min after ABX stimulation. ABX gradually increased the ratio of the CBF and CBA (normalized CBF and CBA) by 30% within 10 min, and the ratios of the CBF and CBA at 10 min after stimulation were 1.25 (*n* = 13) and 1.27 (*n* = 8), respectively (Figure 1A). We examined the effects of Ca^2+^ on the CBF and CBA stimulated by ABX. The switch to a nominally Ca^2+^-free solution decreased the ratios of CBF and CBA by 5% within 5 min, and then stimulation using ABX increased the ratios of the CBF and CBA by 10–13% within 5 min. The ratios of the CBF and CBA at 5 min after the switch were 0.96 (*n* = 8) and 0.96 (*n* = 6) and those 5 min after the addition of ABX were 1.04 and 1.09, respectively (Figure 1B). The same experiments were carried out using nifedipine (20 µM) instead of the Ca^2+^-free solution. The addition of nifedipine decreased the ratios of the CBF and CBA by 5% and then ABX stimulation increased them by 10%. A previous study has shown similar results in c-LAECs [6]. Experiments were also carried out in the absence of CO_2_/HCO_3_^−^ (Figure 1C,D). The switch to the CO_2_/HCO_3_^−^-free control solution immediately increased the CBF and CBA, and the ratios of the CBF and CBA 5 min after the switch were 1.32 (*n* = 8) and 1.21 (*n* = 6), respectively. Further ABX stimulation increased only the CBA, but not the CBF. The ratios of the CBF and CBA at 5 min after ABX stimulation were 1.32 (*n* = 6) and 1.30 (*n* = 8), respectively. The effects of nifedipine (10 µM) on the CBA increase were examined (Figure 1D). The addition of nifedipine decreased the CBF and CBA by 5%. Then, the switch to a CO_2_/HCO_3_^−^-free solution immediately increased the CBF and CBA. The ratios of the CBF and CBA 5 min after the switch were 1.28 (*n* = 4) and 1.20 (*n* = 4), respectively. Further ABX stimulation did not increase the CBA (Figure 1D). Similar results were obtained using a Ca^2+^-free solution. The CBA increase stimulated by ABX in the application of the CO_2_/HCO_3_^−^-free solution appears to be controlled by a [Ca^2+^]_i_ increase via the Ca_V_1.2 activation.

Prior addition of BAPTA-AM (10 µM, a membrane permeable analog of BAPTA, a Ca^2+^ chelator, that binds the intracellular calcium after the acetoxymethyl group is removed by the cytoplasmic esterase) completely inhibited the increases in the CBF and CBA stimulated by ABX in a Ca^2+^-free solution [6]. Thus, Ca^2+^ entry via the ABX-activated Ca_V_1.2 triggers two signal pathways, CO_2_/HCO_3_^−^-dependent and CO_2_/HCO_3_^−^-independent pathways.

#### 2.1.2. ABX-Stimulated Increases in [Ca^2+^]_i_

Changes in [Ca^2+^]_i_ were monitored via the fura-2 fluorescence ratio (F340/F380) in the c-LAECs. In the control solution, ABX stimulation gradually increased the F340/F380, which reached a plateau within 15 min, and then, the addition of nifedipine decreased the F340/F380 to the pre-stimulation level within 10 min (Figure 2A). Changes in the F340/F380 stimulated by ABX were measured in the c-LAECs perfused with a Ca^2+^-free solution. ABX induced a small transient increase in the F340/F380 (Figure 2B), as shown in a previous report [6]. ABX has been shown to stimulate the Ca^2+^ release from acidic stores in alveolar type II cells (ATII cells), which was inhibited by an increase in pH_i_ [6,13]. Experiments were then carried out in a CO_2_/HCO_3_^−^-free solution. The switch from the control solution to the CO_2_/HCO_3_^−^-free solution did not change the F340/F380, and ABX stimulation did not increase the F340/F380. The addition of nifedipine did not change the F340/F380 (Figure 2C). An increase in pH_i_ induced by the application of the CO_2_/HCO_3_^−^-free solution inhibited ABX-stimulated Ca^2+^ release from the acidic stores, as shown in a previous report [6]. Moreover, the CO_2_/HCO_3_^−^-free solution may hyperpolarize the membrane potential by inhibiting the NBC [6], which decreases the [Ca^2+^]_i_ to a low level.

#### 2.1.3. ABX-Stimulated Changes in pH_i_

Changes in pH_i_ were measured via the ratio of SNARF1 fluorescence (F645/F592). In the control solution, ABX stimulation gradually increased the pH_i_ from 7.49 to 7.65 (*n* = 8) within 15 min (Figure 3A). The cells were also stimulated by ABX in a Ca^2+^-free solution (Figure 3B) or in the presence of nifedipine (Figure 3C). ABX increased the pH_i_ from 7.51 to 7.58 in the Ca^2+^-free solution (*n* = 4, Figure 3B) and from 7.52 to 7.60 in the presence of nifedipine (*n* = 5, Figure 3C). Thus, the activation of Ca_V_1.2 significantly increased the pH_i_ in the ABX-stimulated c-LAECs. However, the Ca^2+^ release from the acidic stores also increased the pH_i_, but its extent was small (Figure 3B,C). Changes in pH_i_ were measured upon applying the CO_2_/HCO_3_^−^-free solution. An application of the CO_2_/HCO_3_^−^-free solution transiently increased and sustained the pH_i_. The further ABX stimulation did not change the pH_i_ (Figure 3D).

#### 2.1.4. ABX-Stimulated Cell Shrinkage and a Decrease in [Cl^−^]_i_

##### Video Images of ABX-Stimulated Cell Shrinkage and Enhancement of MQAE Fluorescence in c-LAECs

Figure 4A shows a phase contrast video image of a typical c-LAEC perfused with the control solution before ABX stimulation (Figure 4A) and that at 15 min after ABX stimulation (Figure 4B). The outline of a c-LAEC before ABX stimulation is shown in Figure 4A and was superimposed in Figure 4B. The c-LAECs stimulated by ABX were smaller than those before ABX stimulation, indicating that ABX decreased the cell volume. Cell shrinkage is known to decrease [Cl^−^]_i_ [9,12]. We monitored the [Cl^−^]_i_ using a Cl^−^-sensitive fluorescence dye in the c-LAECs (MQAE) [9,12]. ABX stimulation increased the intensity of MQAE fluorescence in one of the c-LAECs (Figure 4C,D), indicating that ABX stimulation decreases [Cl^−^]_i_.

##### ABX-Stimulated Decreases in Cell Volume and [Cl^−^]_i_

In the control solution, the changes in cell volume (V/V_0_, index of cell volume) and [Cl^−^]_i_ (F_0_/F, MQAE fluorescence ratio) stimulated by ABX were measured in the c-LAECs (Figure 5). ABX stimulation decreased the V/V_0_ to 0.81 (*n* = 5, 6 min after ABX stimulation) (Figure 5A) and the F_0_/F to 0.70 (*n* = 4, 10 min after the ABX stimulation) (Figure 5B). Then, the addition of nifedipine immediately recovered the V/V_0_ to the level before ABX stimulation (V/V_0_ at 5 min after nifedipine addition = 1.03) (Figure 5A) Thus, ABX stimulation decreased the cell volume (V/V_0_) and MQAE fluorescence ratio (F_0_/F) by activating Ca_V_1.2.

To examine the relationship between the CBA and V/V_0_ (or [Cl^−^]_i_) without a pH_i_ change, experiments were also carried out using the CO_2_/HCO_3_^−^-free solution, in which ABX stimulation increased only the CBA. The switch to the CO_2_/HCO_3_^−^-free solution decreased the V/V_0_ and F_0_/F to 0.90 (*n* = 5, 5 min after the switch) and 0.84 (*n* = 4, 5 min after the switch), respectively. Further ABX stimulation decreased the V/V_0_ and F_0_/F to 0.78 (*n* = 5, 10 min after ABX stimulation, Figure 5C) and 0.69 (10 min after ABX stimulation, Figure 5D), respectively. Thus, in the CO_2_/HCO_3_^−^-free solution, ABX still induced cell shrinkage, leading to [Cl^−^]_i_ decrease, although ABX did not increase the [Ca^2+^]_i_ (Figure 2C). The question is whether or not the cell shrinkage or [Cl^−^]_i_ decrease was Ca_V_1.2-dependent. The effects of nifedipine on ABX-stimulated cell shrinkage and [Cl^−^]_i_ decrease were examined in the CO_2_/HCO_3_^−^-free solution (Figure 5E,F). In the control solution, the addition of nifedipine increased the V/V_0_ and F_0_/F to 1.12 (*n* = 4, 5 min after nifedipine addition) and 1.15 (*n* = 5, 5 min after nifedipine addition), respectively. The switch to the CO_2_/HCO_3_^−^-free solution decreased the V/V_0_ and F_0_/F to 0.93 and 0.82, and then ABX stimulation did not change the V/V_0_ (0.91 at 10 min after ABX stimulation) (Figure 5E) and F_0_/F (0.82 at 10 min after ABX stimulation) (Figure 5F). The ABX-stimulated cell shrinkage and [Cl^−^]_i_ decrease in the CO_2_/HCO_3_^−^-free solution are likely to be induced by the activation of Ca_V_1.2.

### 2.2. CO_2_/HCO_3_^−^-Dependent Pathway (pH_i_ Pathway)

The pH_i_ pathway (pH_i_ increase) is activated by the HCO_3_^−^ entry. The epithelial cells in airways express two HCO_3_^−^ transporters, Na^+^-HCO_3_^−^ cotransport (NBC) and Cl^−^/HCO_3_^−^ exchange (anion exchange, AE) [17].

#### 2.2.1. Effects of DIDS on the CBF, CBA and pH_i_

The effects of 4,4′-Diisothiocyano-2,2′-stilbenedisulfonic acid (DIDS, 100 µM, an inhibitor of the NBC and AE) on the CBF, CBA and pH_i_ were examined (Figure 6). The addition of DIDS (100 µM) gradually increased the CBF, but not the CBA (Figure 6A). The CBF reached a plateau within 15 min. The CBF and CBA ratios 15 min after DIDS addition were 1.12 (*n* = 6) and 1.01 (*n* = 4), respectively. The addition of DIDS also gradually increased the pH_i_ from 7.45 to 7.52 (*n* = 5, 10 min after DIDS addition), and ABX stimulation did not affect the gradual pH_i_ increase induced by DIDS (7.55 (*n* = 4), 10 min after ABX stimulation, Figure 6B). The effects of ABX on the CBA and CBF were examined in the presence of DIDS (Figure 6C). The addition of DIDS gradually increased only the CBF. Then, stimulation using ABX immediately increased only the CBA by 17% without any increase in the CBF (Figure 6C). In the DIDS-treated c-LAECs, ABX did not increase the CBF and pH_i_, but it increased the CBA (Figure 6C). Changes in the MQAE fluorescence ratio induced by ABX were measured in the presence of DIDS. DIDS alone did not change the F_0_/F, but stimulation with ABX decreased the F_0_/F (Figure 6D). DIDS did not affect the [Cl^−^]_i_ decrease stimulated by ABX, suggesting that a [Cl^−^]_i_ decrease increased the CBA.

#### 2.2.2. Effects of HCO_3_-Containing NO_3_^−^ Solution on CBF, CBA and pH_i_

To confirm HCO_3_^−^ entry via the NBC, we used a HCO_3_^−^-containing Cl^−^-free NO_3_^−^ solution, in which the NBC is functional, but not the AE (Figure 6E,F). The switch to the HCO_3_^−^-containing Cl^−^-free NO_3_^−^ solution immediately increased the CBF and CBA, and then stimulation using ABX increased both. The CBF and CBA ratios at 10 min after the switch were 1.24 (*n* = 7) and 1.20 (*n* = 8), and those at 10 min after the ABX stimulation were 1.33 and 1.33, respectively (Figure 6E). The HCO_3_^−^-containing Cl^−^-free NO_3_^−^ solution alone increased the pH_i_ from 7.43 to 7.65 (*n* = 5) and then, the ABX stimulation increased the pH_i_ to 7.91. The HCO_3_^−^-containing Cl^−^-free NO_3_^−^ solution potentiated the CBF, CBA and pH_i_ being increased by ABX (Figure 6F). The HCO_3_^−^-containing Cl^−^-free NO_3_^−^ solution, which inhibits HCO_3_^−^ extrusion via the AE while maintaining HCO_3_^−^ influx via the NBC, may increase the concentration of HCO_3_^−^ in the c-LAECs, leading to an increase in pH_i_. ABX appears to stimulate the NBC to increase the pH_i_.

### 2.3. CO_2_/HCO_3_^−^-Independent Pathway (Cl^−^ Pathway)

An increase in the CBA appears to be activated by an [Cl^−^]_i_ decrease (Cl^−^ pathway), coupled with cell shrinkage [12]. To inhibit the [Cl^−^]_i_ decrease, we used inhibitors of the Cl^−^ channel, 5-nitro-2-(3-phenylpropylamino)benzoic acid (NPPB, 20 µM) and T16Ainh (10 µM, an inhibitor of ANO1). A previous study demonstrated that anoctamin-1 (ANO1), a Ca^2+^-activated Cl^−^ channel, functions in ciliated nasal epithelial cells [12,18].

#### 2.3.1. Effects of NPPB on CBF, CBA and [Cl^−^]_i_

The addition of NPPB decreased the CBF and CBA ratios to 0.92 and 0.96 within 5 min, respectively. Then, the ABX stimulation immediately increased the CBF and CBA ratios to 1.06 and 1.10, respectively (Figure 7A). The addition of NPPB increased the MQAE fluorescence ratio (F_0_/F) from 0.99 (*n* = 6) to 1.14 (5 min after the addition). Then, the ABX stimulation gradually increased the F_0_/F (F_0_/F at 15 min after ABX stimulation = 1.25) (Figure 7B). In the experiments using the CO_2_/HCO_3_^−^-free solution, NPPB did not affect the increases in the CBF and CBA induced by the switch to the CO_2_/HCO_3_^−^-free solution. However, ABX stimulation did not increase the CBF or CBA (Figure 7C). The CBF and CBA ratios before and 15 min after ABX stimulation were 1.03 (*n* = 4) and 1.11 (*n* = 6) and 1.03 and 1.06 in the CO_2_/HCO_3_^−^-free solution with NPPB, respectively. Changes in [Cl^−^]_i_ were monitored via the MQAE fluorescence ratio (F_0_/F) (Figure 7D). The addition of NPPB increased the F_0_/F by 15%. The switch to a CO_2_/HCO_3_^−^-free solution, in which there was decreased Na^+^ entry due to inhibition of the NBC, decreased the F_0_/F to 0.88 (*n* = 5) at 5 min after the switch, and stimulation with ABX did not change the F_0_/F (0.86 at 15 min after ABX stimulation). An increase in [Cl^−^]_i_ appears to inhibit the CBF and CBA increase stimulated by ABX [12]. No decrease in [Cl^−^]_i_ induced no further increase in the CBA during ABX stimulation in the CO_2_/HCO_3_^−^-free solution.

#### 2.3.2. Effects of an ANO1 Inhibitor (T16Ainh) on CBF, CBA and [Cl^−^]_i_

The addition of T16Ainh (10 µM, an inhibitor of ANO1) decreased the CBA (0.95 (*n* = 5) at 5 min after the addition), but not the CBF (1.02 (*n* = 7) at 5 min after the addition). ABX stimulation increased the CBF and CBA to 1.12 and 1.12 (at 10 min after the ABX stimulation), respectively (Figure 8A). Experiments were then carried out in a CO_2_/HCO_3_^−^-free solution. T16Ainh did not affect the CBF and CBA increased by the switch to the CO_2_/HCO_3_^−^-free solution. Then, further ABX stimulation did not change the CBF or CBA. The CBF and CBA ratios before and after ABX stimulation were 1.23 (*n* = 7) and 1.18 (*n* = 5) at 5 min after the T16Ainh addition and 1.23 and 1.20 at 10 min after the ABX stimulation, respectively (Figure 8B). These results indicate that the activation of ANO1 decreases the [Cl^−^]_i_ in ABX-stimulated c-LAECs.

### 2.4. Effects of a CO_2_/HCO_3_^−^-Free Cl^−^-Free NO_3_^−^ Solution on CBF, CBA and [Cl^−^]_i_

These results suggest that an increase in pH_i_ and a decrease in [Cl^−^]_i_ stimulated by ABX increased the CBF and CBA. We used a CO_2_/HCO_3_^−^-free Cl^−^-free NO_3_^−^ solution, which increases the pH_i_ and decreases the [Cl^−^]_i_ [9]. The switch to the CO_2_/HCO_3_^−^-free solution increased the CBF and CBA. The CBF and CBA ratios at 5 min after the switch were 1.25 (*n* = 5) and 1.16 (*n* = 5), respectively. Then, the second switch to the CO_2_/HCO_3_^−^-free Cl^−^-free NO_3_^−^ solution increased the CBA without any CBF increase. The CBF and CBA ratios at 5 min after the second switch were 1.27 (*n* = 5) and 1.28 (*n* = 5), respectively. Further ABX stimulation did not increase the CBA or CBF. The CBF and CBA ratios at 5 min after ABX stimulation were 1.31 (*n* = 5) and 1.29 (*n* = 5), respectively (Figure 9A). Changes in [Cl^−^]_i_ were also monitored via the MQAE fluorescence ratio using the same protocol. The CO_2_/HCO_3_^−^-free solution decreased the F_0_/F to 0.90 (5 min after the switch, *n* = 4) and the second switch further decreased the F_0_/F to 0.73 (5 min after the second switch, *n* = 4). ABX stimulation did not decrease the F_0_/F (0.69 at 10 min after the ABX stimulation, *n* = 4) (Figure 9B). Thus, the effects of ABX on the CBA and CBF were mimicked by the CO_2_/HCO_3_^−^-free Cl^−^-free NO_3_^−^ solution.

### 2.5. Expression of Anoctamin-1 in c-LAECs

The expression of ANO1 (TMEM16A) was examined using Western blotting and immunofluorescence. In the Western blotting, a single band of ANO1 was detected at 110 kDa (Figure 10). The immunofluorescence analysis of ANO1 revealed that the cilia and cell bodies were positively stained for ANO1 in the c-LAECs (Figure 11A), and the cilia were positively stained for acetylated tubulin (α-tubulin, a marker of cilia) (Figure 11B). The merged image shows that ANO1 exists in the cilia (Figure 11C). A phase contrast image of the c-LAECs is shown in Figure 11D.

## 3. Discussion

The activation of Ca_V_1.2, a voltage-dependent Ca^2+^ channel, maintained the sustained increase in [Ca^2+^]_i_ in ABX-stimulated-cLAECs, in which the contribution of Ca^2+^ release from the stores was small [6]. This study demonstrated that the [Ca^2+^]_i_ increase using ABX triggers CBF and CBA increases in the c-LAECs by activating two signaling pathways: the pH pathway (an increase in pH_i_) and the Cl^−^ pathway (a decrease in [Cl^−^]_i_). The pH pathway increased the CBF by 30% and the CBA by 15–20%, and the Cl^−^ pathway increased the CBA by 10–15%.

The [Ca^2+^]_i_ increase is essential to increase the CBF and CBA in the ABX-stimulated c-LAECs, since prior treatment with BAPTA-AM abolished their increases. This suggests that the non-specific actions of ABX, which are not mediated by an [Ca^2+^]_i_ increase, are negligibly small in the regulation of the CBF and CBA. DIDS, which inhibited the pH pathway, did not affect the ABX-stimulated increase in [Ca^2+^]_i_, since ABX stimulation increased only the CBA, but not the CBF, although the [Ca^2+^]_i_ increase appears to be sufficient to increase the CBF. Moreover, in the presence of NPPB or T116Ainh, ABX stimulation induced small increases in the CBF and CBA, suggesting the activation of the pH pathway. ABX is likely to increase [Ca^2+^]_i_ to a sufficient level to activate the pH pathway. A [Cl^−^]_i_ increase on the part of the inhibitors of the Cl^−^ channels appears to decrease the CBF and CBA enhanced by the pH pathway. Yasuda et al. showed that an increase in [Cl^−^]_i_ decreases the CBF and CBA [12]. Moreover, in the CO_2_/HCO_3_^−^-free solution, ABX increased only the CBA and decreased the cell volume and [Cl^−^]_i_, effects that were eliminated using a Ca^2+^-free solution or nifedipine. In a CO_2_/HCO_3_^−^-free solution, ABX may increase the [Ca^2+^]_i_ to a level activating ANO1 in the cilia. Saito et al. suggest that ABX increases the [Ca^2+^]_i_ only in the cilia, although no increase in the [Ca^2+^]_i_ of the cell body was detected in the CO_2_/HCO_3_^−^-free solution [6]. In the olfactory cilia, the [Ca^2+^]_i_ increase is limited in the same vicinity for a long time [19]. Moreover, we measured the [Ca^2+^]_i_ in c-LAECs stimulated by ABX and ionomycin (IM, 1 µM), using fluo4 fluorescence. The increases in the fluorescence ratio (F/F_0_) stimulated by ABX were 10–15% of those stimulated by IM (1 µM) (Appendix A). The direct activation of the CBF by a [Ca^2+^]_i_ increase has been established in the beating cilia of the airways [3,4,20]. However, in the ABX-stimulated c-LAECs, increases in [Ca^2+^]_i_ may have been too small to directly increase the CBF and CBA.

An increase in pH_i_ enhances the CBF and CBA in airway ciliary cells [9,10] and enhances the CBF in sperm flagella [21,22]. In the c-LAECs, the pH pathway is activated by HCO_3_^−^ entry via the NBC, which is inhibited by DIDS. There are two bicarbonate transporters in the c-LAECs: the NBC and AE [17]. The DIDS-sensitive AE exists in the apical membrane and mediates HCO_3_^−^ secretion in the bronchiole epithelial cells [17]. Moreover, in the HCO_3_^−^-containing Cl^−^-free solution, ABX enhanced the increase in pH_i_, indicating that the AE does not transport HCO_3_^−^ into the cell in c-LAECs. In this solution, no HCO_3_^−^ secretion via the AE occurs because of no extracellular Cl^−^. The inhibition of AE (no HCO_3_^−^ extrusion via the AE retaining HCO_3_^−^ uptake via the NBC) increases the intracellular HCO_3_^−^ concentration to elevate the pH_i_. Based on these results, we concluded that ABX stimulated the NBC, mediated by a [Ca^2+^]_i_ increase, leading to an increase in pH_i_. Fois et al. demonstrated that ABX increases pH_i_ in type II pneumocytes [13]. The messenger RNAs of all NBC isoforms are expressed in the airway epithelial cells [17], but the membrane localization of the NBC isoforms has not been identified, although NBCe1 and NBCe2 have been identified in the basolateral membrane of Calu-3 cells [23]. Moreover, DIDS did not affect the decrease in [Cl^−^]_i_, indicating that the [Cl^−^]_i_ decrease is not induced by DIDS-sensitive Cl^−^ channels in the c-LAECs.

The pH pathway was still activated by ABX in the presence of nifedipine or in a Ca^2+^-free solution. This finding indicates that the ABX-stimulated small transient Ca^2+^ release from the acidic stores activates the NBC in the c-LAECs, leading to a pH_i_ increase. Lieb et al. reported that a transient [Ca^2+^]_i_ increase induced by short-term ATP stimulation has been shown to induce a prolonged CBF increase in human airway ciliary cells, and they suggested that a transient [Ca^2+^]_i_ increase phosphorylates target proteins to induce a prolonged CBF increase [20]. The ABX-stimulated small transient [Ca^2+^]_i_ increase appears to phosphorylate target proteins, including the NBC, to activate the pH pathway.

This study also demonstrated that a pH_i_ increase enhances not only the CBF but also the CBA in the c-LAECs. Cilia have two functionally distinct molecular motors: the ODAs and the IDAs. The ODAs control the CBF and the IDAs control the waveform, including the CBA [15]. An increase in the pH_i_ is suggested to directly act on the ODAs in sperm flagella [21,22]. Although there is no report showing that pH_i_ affects the IDAs, a pH_i_ increase may activate the IDAs, in a similar mechanism to activating the ODAs, to increase the CBA. The activation mechanisms of ODAs or IDAs due to a pH_i_ increase remain uncertain. Previous reports have suggested that pH_i_-induced changes in the histidine charge may affect the activity of dynein ATPase [24]. A study on sperm flagella suggests that an increase in the pH_i_ activates dynein via the pH-dependent and cAMP-independent phosphorylation of dynein components and/or other axonemal proteins [22].

Previous studies have demonstrated that a decrease in [Cl^−^]_i_ enhances the CBA at 37 °C, but not the CBF [9,12]. [Cl^−^]_i_ is decreased by the cell shrinkage under iso-osmotic conditions [12]. ABX stimulated a cell shrinkage and a [Cl^−^]_i_ decrease in c-LAECs. Previous studies have demonstrated that Ca_V_1.2 is expressed in the cilia and is activated by ABX [6]. The Ca^2+^ influx via Ca_V_1.2 activated the Cl^−^ pathway in the c-LAECs, but the Ca^2+^ release from the acidic stores had little effect on the Cl^−^ pathway, because ABX still decreases [Cl^−^]_i_ without a Ca^2+^ release in a CO_2_/HCO_3_^−^-free solution.

The Cl^−^ pathway, which was independent of HCO_3_^−^ (pH_i_ pathway), was activated by a Cl^−^-free NO_3_^−^ solution and inhibited by Cl^−^ channel blockers (NPPB and T16Ainh) in the c-LAECs. This study demonstrated that ANO1 (a Ca^2+^-activated Cl^−^ channel [18]) is activated by Ca_v_1.2. Moreover, both channels are expressed in the cilia of the c-LAECs [6,12]. The coupling of ANO1 and Ca_V_1.2 in the cilia plays a key role in activating the Cl^−^ pathway in ABX-stimulated c-LAECs. However, ANO1 was also expressed in the cell body. The ANO1 expressed in the cell body (ANO1_cell body_) may be activated by the [Ca^2+^]_i_ increase caused by Ca_V_1.2 during ABX stimulation. The activation of ANO1 in the cell body stimulates the Cl^−^ release from the basolateral membrane and enhances cell shrinkage in c-LAECs. However, increases in [Ca^2+^]_i_ are small in the cell body during ABX stimulation (Appendix A). In a previous report, no [Ca^2+^]_i_ increase was detected in an airway cell line during ABX stimulation [14]. Moreover, in this study, no [Ca^2+^]_i_ increase was detected in the CO_2_/HCO_3_^−^-free solution [6]. However, experiments using nifedipine or a Ca^2+^-free solution revealed that ABX still activated both Ca_V_1.2 and ANO1 in this solution. In the cilia, the activation of Ca_V_1.2 may increase [Ca^2+^]_i_ to a significant level, activating ANO1 in the CO_2_/HCO_3_^−^-free solution. Based on these observations, the activities of ANO1 in the cell body may be low in ABX-stimulated c-LAECs, although we cannot neglect the contribution of the ANO1 in the cell body to decreased [Cl^−^]_i_.

The Cl^−^ pathway increased the CBA, but not the CBF. However, an increase in [Cl^−^]_i_ induced by NPPB decreased both the CBF and CBA. Thus, the effects of [Cl^−^]_i_ on the CBA and CBF are different. Yasuda et al. suggested that the concentration–response curve of the CBA to [Cl^−^]_i_ shifts to a lower concentration than that of the CBF [12]. A level of [Cl^−^]_i_ under the control conditions may maintain the IDA activity, controlling the CBA at the lowest level, but it may maintain the ODA activity, controlling the CBF at the highest level.

There are many reports showing that decreased [Cl^−^]_i_ enhances cellular functions in many cell types, including the airway ciliary cells [25,26,27,28,29]. These observations suggest that a Cl^−^ sensor exists in these cells. With-no-lysine kinase1 (WNK1) and WNK4 have been shown to be intracellular Cl^−^ sensors [27,28,29]. Piala et al. demonstrated that the Cl^−^ binds to the kinase domain of WNK1 and suppresses its activity by inhibiting autophosphorylation [28]. The actions of WNK1 and WNK4 were studied in the NaCl cotransporter of distal nephrons [27,28,29]. There are four subtypes of WNK, WNK1-4, and the chloride binding site of the kinase domain is conserved among them, but the activities of the WNK subtypes are inhibited by different Cl^−^ concentrations, that is, WNK1 by 60–150 mM, WNK3 by 100–150 mM and WNK4 by 0–40 mM [27,28]. WNK1 and WNK4 have been shown to regulate epithelial Na^+^ channels [30,31], CFTR [32], NKCC1 [33] and NCC [26,34]. Based on these observations, our hypothesis is that WNK1 controls the ODAs and WNK4 controls the IDAs in the c-LAECs. Further experiments are needed to verify this hypothesis.

ABX at a high concentration, such as 100 µM, inhibits voltage-gated Na^+^ channels (Na_V_s) [35]. ABX at 1 µM is unlikely to inhibit Na_V_s. Ciliated LAECs express epithelial Na^+^ channels, the activation of which induces a dry airway surface by accelerating the fluid absorption observed in cystic fibrosis patients. An increase in Na^+^ influx induces cell swelling to increase [Cl^−^]_i_. An increase in [Cl^−^]_i_ decreases the CBF and CBA [12]. In healthy airways, the c-LAECs secrete Cl^−^ [14]. The impairment of Cl^−^ secretion, such as in cystic fibrosis, may increase [Na^+^]_I_, leading to an [Cl^−^]_i_ increase, which may decrease the CBF and CBA [12]. The inhibition of the Na_V_s using 10 µM ABX appears to have little effect on the CBF and CBA in c-LAECs.

Figure 12 shows a schematic diagram of the ABX-stimulated c-LAECs. ABX increases [Ca^2+^]_i_ by stimulating the nifedipine-sensitive Ca_V_1.2 in the c-LAECs. The increases in [Ca^2+^]_i_ activate two signaling pathways: the pH_i_ pathway and the Cl^−^ pathway. The activation of Cav1.2 maintains most of the increase in [Ca^2+^]_i_. The [Ca^2+^]_i_ increase activates NBC-entering HCO_3_^−^ to increase the pH_i_. The pH_i_ pathway (an elevation in pH_i_) increases the CBF and CBA by 30% and 15–20%, respectively, in ABX-stimulated c-LAECs. The [Ca^2+^]_i_ increase also activates the ANO1 existing in the cilia to accelerate Cl^−^ release, leading to the decrease in [Cl^−^]_i_, and may also activate the ANO1 existing the cell body. The Cl^−^ pathway (an [Cl^−^]_i_ decrease) enhances the CBA by 10–15%, and was completely inhibited by nifedipine or T16Ainh. The Ca^2+^ release from the acidic stores, which is small and transient, also increases [Ca^2+^]_i_ and activates the NBC in the c-LAECs.

## 4. Materials and Methods

### 4.1. Ethical Approval

The experiments were approved by the Committees for Animal Research of Kyoto Prefectural University of Medicine (No. 26-263, April 2017) and Ritsumeikan University (BKC-HM-2017-050). The animals were cared for and the experiments were carried out according to the guidelines of these committees. Female mice (C57BL/6J, 5 weeks of age) were purchased from Shimizu Experimental Animals (Kyoto, Japan), fed standard pellet food and water ad libitum and were used for experiments at 6–10 weeks of age. The mice were first anaesthetized using inhalational isoflurane (3%) and were further anesthetized using an intraperitoneal injection (ip) of pentobarbital sodium (70 mg/kg) and heparinized (1000 units/kg) for 15 min. Then, the mice were sacrificed with a high dose of pentobarbital sodium (100 mg/kg, ip).

### 4.2. Solutions and Chemicals

The CO_2_/HCO_3_^−^-containing control solution (control solution) contained (in mM): NaCl, 121; KCl, 4.5; NaHCO_3_, 25; MgCl_2_, 1; CaCl_2_, 1.5; Na-HEPES, 5; H-HEPES, 5 and glucose, 5. To prepare the CO_2_/HCO_3_^−^-free solution, the HCO_3_^−^ was replaced with Cl^−^. To prepare the Cl^−^-free NO_3_^−^ solution, the Cl^−^ was replaced with NO_3_^−^. To prepare the Ca^2+^-free solution, CaCl_2_ was removed from the solution. The CO_2_/HCO_3_^−^-containing solutions were aerated with 95% O_2_ and 5% CO_2_ and the CO_2_/HCO_3_^−^-free solutions were aerated with 100% O_2_. The pH of the solutions was adjusted to 7.4 by adding 1 N HCl or 1 N HNO_3_, as appropriate. The experiments were carried out at 37 °C. The ABX, nifedipine, NPPB, DIDS and dimethyl sulfoxide (DMSO) were purchased from Sigma (St. Louis, MO, USA), and the heparin, elastase and bovine serum albumin (BSA) were purchased from FUJIFILM Wako (Osaka, Japan). All reagents were dissolved in DMSO and prepared to their final concentrations immediately before the experiments. The DMSO, the concentration of which did not exceed 0.1%, had no effect on the CBF or CBA [6,9,10,11].

### 4.3. Cell Preparation

The ciliated LAECs were isolated from the lungs using an elastase treatment [6,9,11]. Following the elastase treatment, the lungs were minced using fine forceps in a control solution containing DNase I (0.02 mg/mL) and BSA (5%). The minced tissue was filtered through a nylon mesh (a sieve with 300 µm openings). The isolated cells were washed and then suspended in the control solution. The cell suspension was stored at 4 °C and the cells were used within 5 h after the isolation.

### 4.4. CBF and CBA Measurements

The cells were set in a micro perfusion chamber (20 μL) mounted on an inverted light microscope (ECLIPSE Ti, Nikon, Tokyo, Japan) connected to a high-speed camera (Photron Ltd., Tokyo, Japan) and video images were recorded for 2 s at 500 fps (frame per second) [6,9,11]. The stage of the microscope was heated to 37 °C [4]. After the experiments, the CBF and CBA were measured using an image analysis program (DippMotion 2D, Ditect, Tokyo, Japan) [6,9,11]. The normalized CBF and CBA values and CBF and CBA ratios (CBF_t_/CBF_0_ and CBA_t_/CBA_0_), calculated from 4 to 12 cells, were used to make a comparison among the experiments. Each experiment was carried out using 4–10 cover slips with cells obtained from 2 to 5 animals. “n” shows the number of cells.

### 4.5. Measurement of the Cell Volume

The outline of a c-LAEC was traced onto a video image, and the area of the cell (A) was measured using the image analysis program. The index of the cell volume (V_t_/V_0_ = (A_t_/A_0_)^1.5^) was calculated [9]. Each experiment was carried out using 4–6 cover slips obtained from 2 to 3 animals. The V/V_0_ values calculated from 4 to 6 cells were plotted and “n” shows the number of cells.

### 4.6. Measurement of pH_i_, [Cl^−^]_i_ and [Ca^2+^]_i_

Changes in intracellular pH (pH_i_) were monitored via SNARF1 fluorescence (a pH dye) at 37 °C and the cells were set on the heated stage (37 °C) of an inverted confocal laser microscope (model LSM 510-META, Carl Zeiss, Jena, Germany). The excitation was 515 nm and the emissions were 645 nm and 592 nm. The fluorescence ratio (F645/F592) was calculated to measure the pH_i_. The pH_i_ of the c-LAECs was calculated using the calibration line.

Changes in [Cl^−^]_i_ were monitored via MQAE fluorescence (a chloride sensitive dye) [9,12]. The cells were incubated with 10 mM MQAE for 45 min at 37 °C. The MQAE was excited at 780 nm using a 2-photon excitation laser system (Mai Tai^®®^, Spectra-Physics, Santa Clara, CA, USA), and the emission was 510 nm. The ratio of MQAE fluorescence intensity (F_0_/F_t_) was calculated.

Changes in [Ca^2+^]_i_ were monitored via fura-2 fluorescence (a Ca^2+^ dye) as previously reported [6,8]. The ratio of fura-2 fluorescence (F340/F380) was calculated using an image analysis system (MetaFluor, Molecular Devices, San Jose, CA, USA) [6,8].

### 4.7. Western Blotting

The procedure for Western blotting has already been described in previous reports [6]. Protein (5–20 µg) obtained from the isolated lung cells was loaded into each lane. After blocking with 5% skim milk powder, the membrane with the protein was exposed to a primary anti-anoctamin-1 antibody (1:200) (ABN1669 Sigma-Aldrich Merck, Darmstadt, Germany) diluted with solution 1 (Can Get Signal, TOYOBO, Osaka, Japan), and the secondary antibody was applied to the membrane. Then, the antigen–antibody complexes were visualized using a chemiluminescence system (Immobilon, Merck Millipore, Darmstadt, Germany).

### 4.8. Immunofluorescence Examination

The cell suspension containing the isolated lung cells (0.5 mL) was dropped and dried on the coverslip for attaching the cells [6]. After fixation with 4% paraformaldehyde, the cells were stained with an anoctamin-1 antibody and an anti-acetylated tubulin antibody (T6793, Sigma-Aldrich, St. Louis, MO, USA). The cells were washed with PBS containing 0.1% BSA to remove unbound antibodies. Finally, the cells were stained with Alexa 488-conjugated anti-rabbit antibody and Alexa 594-conjugated anti-mouse antibody (Invitrogen, Carlsbad, CA, USA). The cells were observed using confocal laser microscopy (FV10i, Olympus, Tokyo, Japan).

### 4.9. Statistical Analysis

Data are expressed as the means ± SEMs. Statistical significance was assessed using one-way analysis of variance (one-way ANOVA). Differences were considered significant at *p* < 0.05.

## 5. Conclusions

The Ca_V_1.2 existing in the airway cilia plays an important role in the increases in the CBF and CBA in ABX-stimulated c-LAECs. An increase in [Ca^2+^]_i_ induced by the activation of Ca_V_1.2 stimulates two signaling pathways: the pH_i_ pathway and Cl^−^ pathway. This is the first report to clarify the role of Ca_V_1.2 in the activation of beating cilia in the airway. The Ca_V_1.2 existing in cilia, which is activated by some chemicals and receptors, including ABX [6], may be a new target to prevent and improve respiratory problems.

## Figures and Tables

**Figure 1 ijms-24-16976-f001:**
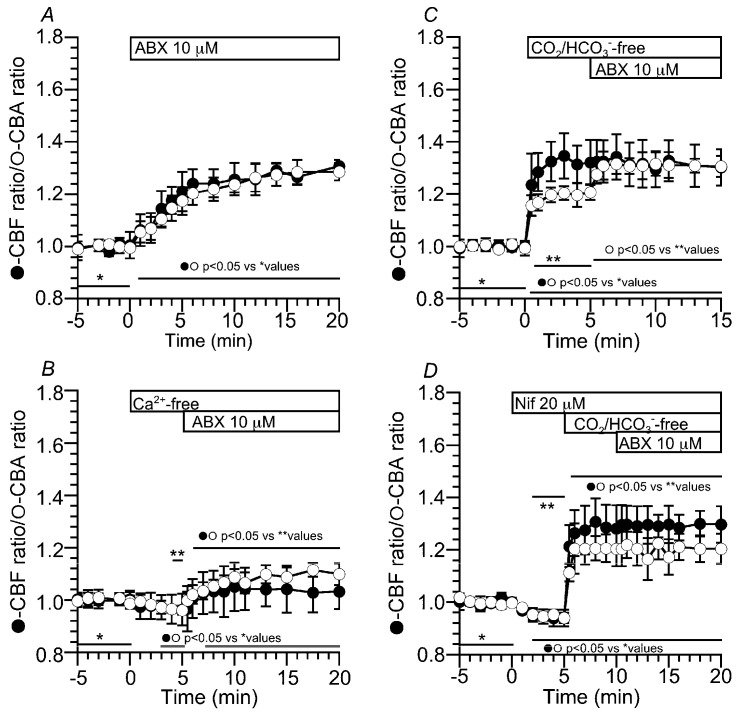
The effect of ABX on the ratios of the CBF and CBA in c-LAECs. (**A**): ABX (10 µM) stimulation in the presence of CO_2_/HCO_3_^−^. In the CO_2_/HCO_3_^−^-containing control solution, ABX stimulation (10 µM) gradually increased the ratios of the CBF and CBA by 30% within 10 min. (**B**): Effects of Ca^2+^-free solution on the ABX-stimulated CBF and CBA in the presence of CO_2_/HCO_3_^−^. The switch to a nominally Ca^2+^-free solution decreased the CBF and CBA ratios by 5% within 5 min. Then, ABX stimulation gradually increased the CBF and CBA ratios by 10%. (**C**): ABX stimulation in the absence of CO_2_/HCO_3_^−^. The switch to the CO_2_/HCO_3_^−^-free solution immediately increased the CBF ratio by 30% and the CBA ratio by 20%. Then, ABX stimulation increased the CBA ratio by 10%, but not the CBF ratio. (**D**): Effects of nifedipine on the ABX-stimulated CBF and CBA ratios in the absence of CO_2_/HCO_3_^−^. The addition of nifedipine (20 µM) decreased the CBF and CBA ratios by 5%. The switch to the CO_2_/HCO_3_^−^-free solution immediately increased the CBF ratio by 30% and the CBA ratio by 20%. Further ABX stimulation did not induce any increase in the CBF ratio or the CBA ratio.

**Figure 2 ijms-24-16976-f002:**
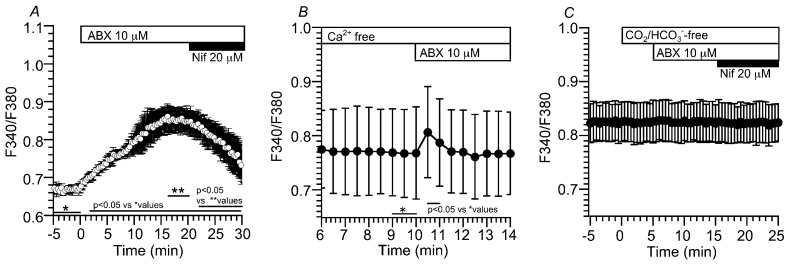
Effects of ABX on [Ca^2+^]_i_ in c-LAECs. Changes in [Ca^2+^]_i_ were monitored via the fura-2 fluorescence ratio (F340/F380). (**A**): ABX stimulation gradually increased F340/F380, which reached a plateau within 15 min. Then, the addition of nifedipine gradually decreased F340/F380 to the control level before ABX stimulation within 10 min. (**B**): Effects of the Ca^2+^-free solution on the ABX-stimulated [Ca^2+^]_i_. In the Ca^2+^-free solution, ABX stimulation induced a small transient increase in F340/F380. (**C**): Effects of ABX on [Ca^2+^]_i_ in the absence of CO_2_/HCO_3_^−^. The switch to the CO_2_/HCO_3_^−^-free solution did not change F340/F380 and then ABX stimulation did not change F340/F380. Further addition of nifedipine did not change F340/F380. The CO_2_/HCO_3_^−^-free solution increased pH_i_, which inhibits Ca^2+^ release from acidic stores, and it inhibits Na^+^ entry via the NBC. The CO_2_/HCO_3_^−^-free solution decreases Ca_V_1.2 channel activity.

**Figure 3 ijms-24-16976-f003:**
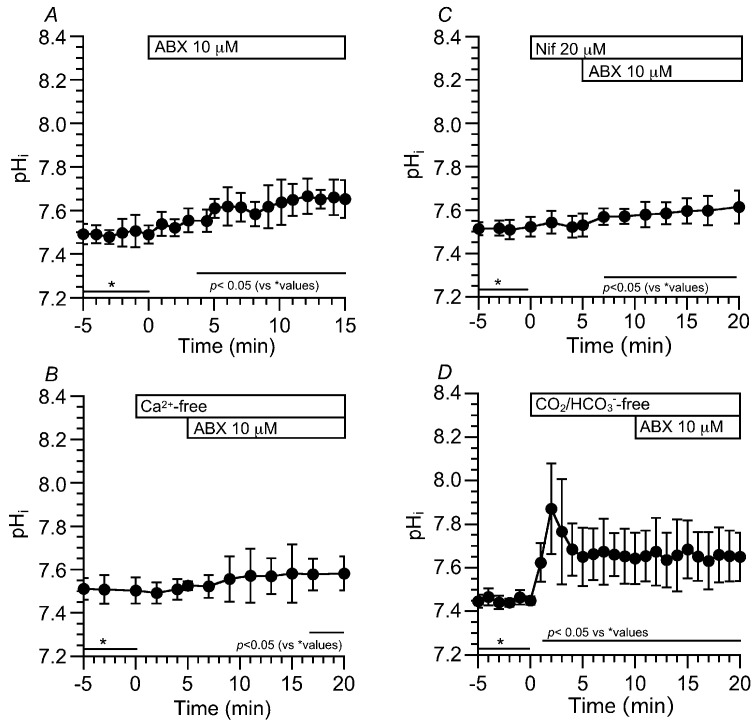
Effects of ABX on pH_i_ in c-LAECs. Changes in pH_i_ were measured using the SNARF1 fluorescence ratio (F645/F592). (**A**): ABX stimulation gradually increased pH_i_ from 7.49 to 7.65 within 15 min (*n* = 8). (**B**): Effects of a Ca^2+^-free solution on the ABX-stimulated pH_i_. The switch to a Ca^2+^-free solution did not change pH_i_. Then, ABX stimulation gradually increased pH_i_ from 7.51 to 7.58 (*n* = 4) (**C**): Effects of nifedipine on the ABX-stimulated pH_i_. The addition of nifedipine did not change pH_i_. Then, ABX stimulation gradually increased pH_i_ from 7.52 to 7.60 (*n* = 5). (**D**): Effects of CO_2_/HCO_3_^−^-free solution on the ABX-stimulated pH_i_. The switch to a CO_2_/HCO_3_^−^-free solution induced a transient increase, followed by a sustained increase in pH_i_. The values of pH_i_ before, 2 min after and 10 min after the switch were 7.45, 7.87 and 7.65 (*n* = 6), respectively. Further ABX stimulation did not change pH_i_.

**Figure 4 ijms-24-16976-f004:**
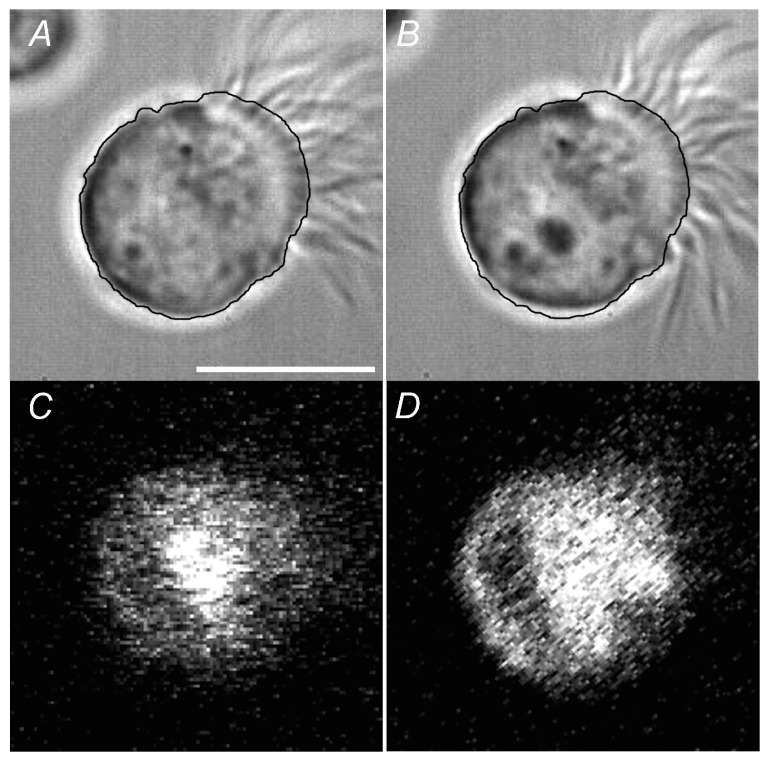
Video frame images and MQAE fluorescence of a c-LAEC. (**A**,**B**): Video frame images of c-LAECs before (**A**) and at 15 min after ABX stimulation (**B**). The outline of a c-LAEC before ABX stimulation is traced by the black line (**A**). ABX stimulation induced cell shrinkage. The traced outline of the c-LAECs in the panel (**A**) was superimposed onto the c-LAEC in the panel (**B**). The panel (**B**) shows that the outline of the c-LAEC stimulated by ABX was smaller than that before ABX stimulation. (**C**,**D**): Changes in MQAE fluorescence of a c-LAEC before (**C**) and at 15 min after ABX stimulation (**D**). ABX stimulation increased the intensity of MQAE fluorescence, indicating that ABX decreased [Cl^−^]_i_ in the c-LAEC.

**Figure 5 ijms-24-16976-f005:**
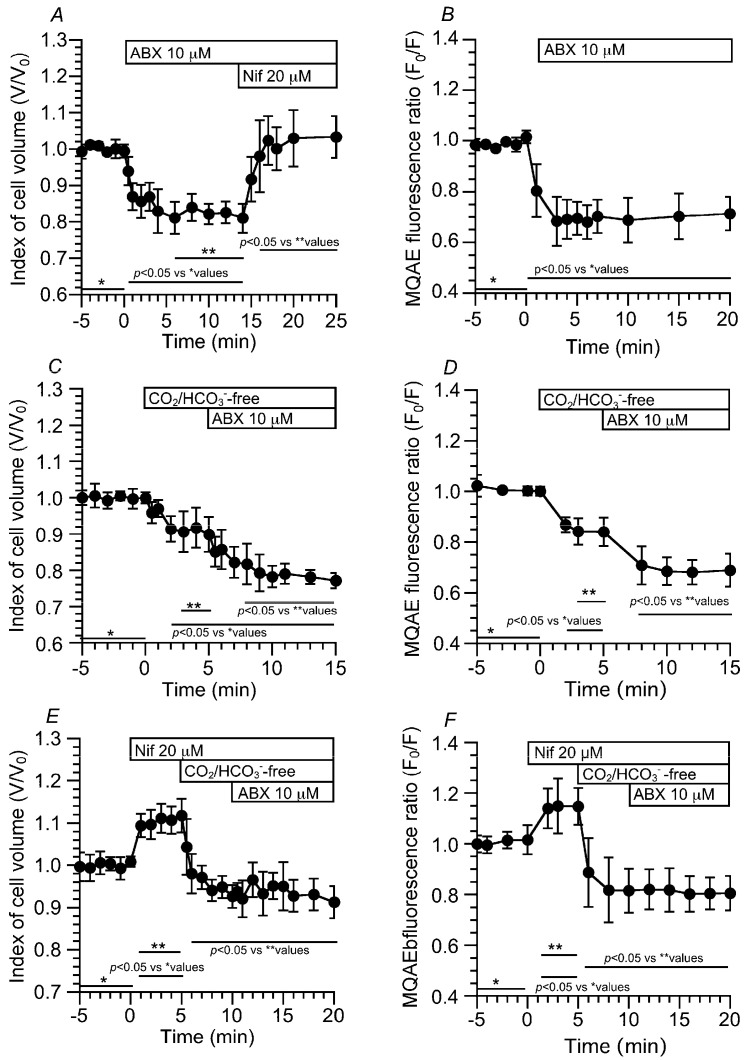
Changes in the cell volume and [Cl^−^]_i_ induced by ABX stimulation in c-LAECs. (**A**): In the CO_2_/HCO_3_^−^-containing control solution, ABX stimulation decreased V/V_0_ by 20%. Then, the addition of nifedipine immediately increased V/V_0_ to the prestimulation level. (**B**): In the CO_2_/HCO_3_^−^-containing control solution, ABX stimulation decreased the ratio of MQAE fluorescence (F_0_/F) by 30%. (**C**): The switch to the CO_2_/HCO_3_^−^-free control solution decreased V/V_0_ by 10%. Then, ABX stimulation gradually decreased V/V_0_. In the CO_2_/HCO_3_^−^-free control solution, ABX still activates Ca_V_1.2 channels. (**D**): The switch to the CO_2_/HCO_3_^−^-free control solution decreased F_0_/F by 15%, and then, ABX stimulation decreased F_0_/F by 20%. (**E**): Effects of nifedipine on V/V_0_ stimulated using ABX. The addition of nifedipine increased V/V_0_ by 10%. Then, the switch to the CO_2_/HCO_3_^−^-free solution decreased V/V_0_ by 20%. Further ABX stimulation did not decrease V/V_0_. (**F**): The addition of nifedipine increased F_0_/F, and the switch to the CO_2_/HCO_3_^−^-free control solution decreased F_0_/F. Further ABX stimulation did not decrease F_0_/F.

**Figure 6 ijms-24-16976-f006:**
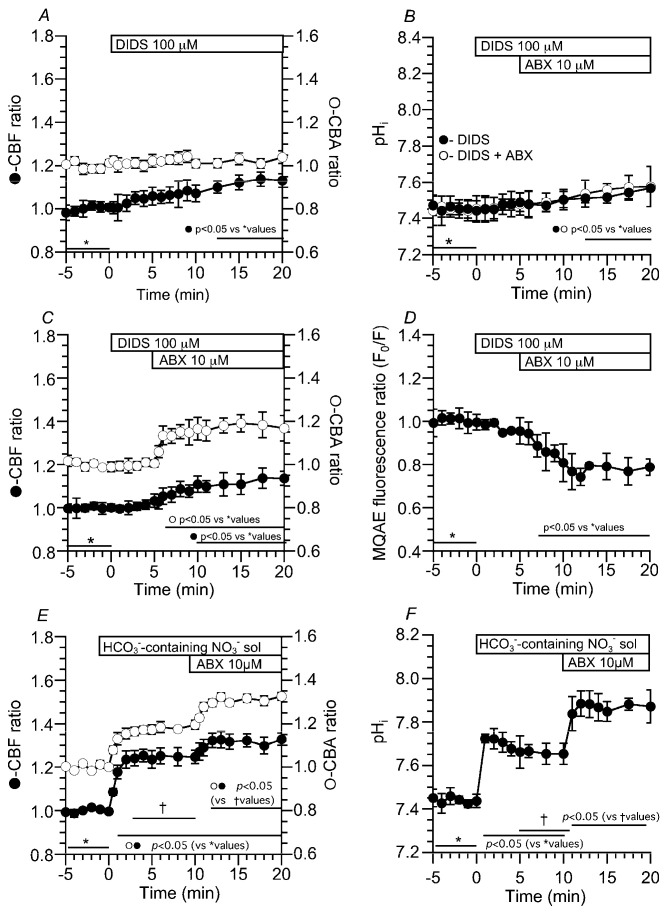
Effects of DIDS on the CBF, CBA, pH_i_ and [Cl^−^]_i_ stimulated by ABX in c-LAECs. (**A**): Effects of DIDS on CBF and CBD. The addition of DIDS gradually decreased the CBF, but not the CBA in c-LAECs. (**B**): Effects of DIDS on pH_i_s with or without ABX. The addition of DIDS gradually increased pH_i_ (J) The ABX stimulation did not affect the gradual pH_i_ increase induced by DIDS (**E**). (**C**): Changes in the CBF and CBA ratios stimulated by ABX in the presence of DIDS. The addition of DIDS gradually increased the CBF ratio by 15%, but not the CBA ratio. Then, ABX stimulation increased the CBA ratio but not the CBF ratio. (**D**): Effects of ABX on [Cl^−^]_i_ of c-LAECs in the presence of DIDS. The addition of DIDS slightly decreased F_0_/F and then ABX stimulation decreased F_0_/F. (**E**): Effects of ABX on the CBF and CBA ratios in the HCO_3_^−^-containing NO_3_^−^ solution. The switch to the HCO_3_^−^-containing NO_3_^−^ solution increased the CBF and CBA ratios. Then, ABX stimulation further increased the CBF and CBA ratios. (**F**): Effects of ABX on pH_i_ in the HCO_3_^−^-containing NO_3_^−^ solution. The switch to the HCO_3_^−^-containing NO_3_^−^ solution increased pH_i_. Then, the ABX stimulation further increased pH_i_. The HCO_3_^−^-containing NO_3_^−^ solution enhanced the pH_i_ increase stimulated by ABX.

**Figure 7 ijms-24-16976-f007:**
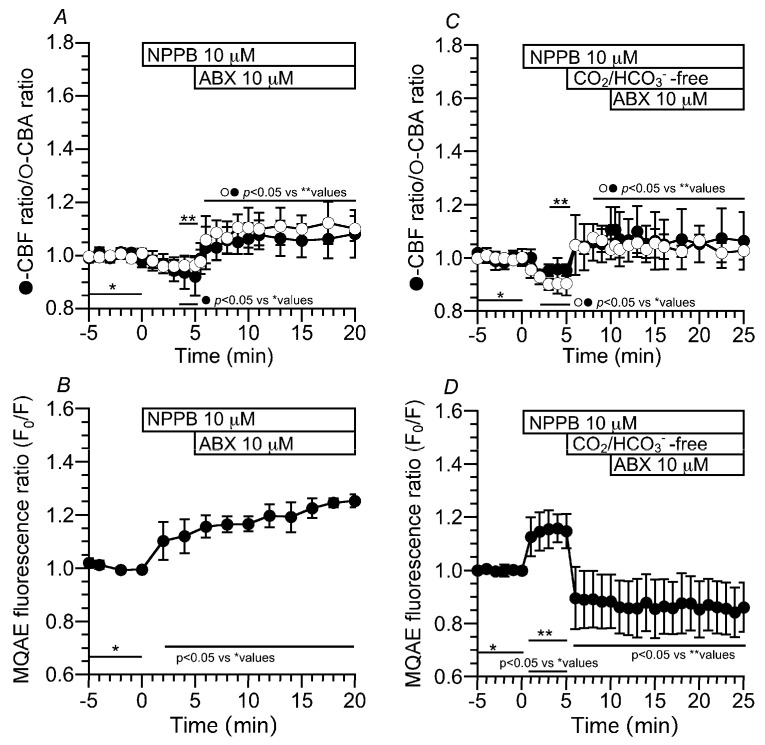
Effects of NPPB (a Cl^−^ channel blocker) on the CBF, CBA and [Cl^−^]_i_ stimulated by ABX. (**A**): In the CO_2_/HCO_3_^−^-containing control solution, the addition of NPPB decreased the CBF and CBA ratios by 5%. Further ABX stimulation increased the CBF and CBA ratios by 10%. (**B**): The addition of NPPB increased F_0_/F. Then, ABX stimulation gradually increased F_0_/F. (**C**): The addition of NPPB decreased the CBF and CBA ratios by 5% and then the switch to the CO_2_/HCO_3_^−^-free solution increased the CBF and CBA ratios. Further ABX stimulation did not change the CBF and CBA ratios. (**D**): The addition of NPPB increased F_0_/F., and then, the switch to the CO_2_/HCO_3_^−^-free control solution decreased F_0_/F. Further ABX stimulation did not change F_0_/F.

**Figure 8 ijms-24-16976-f008:**
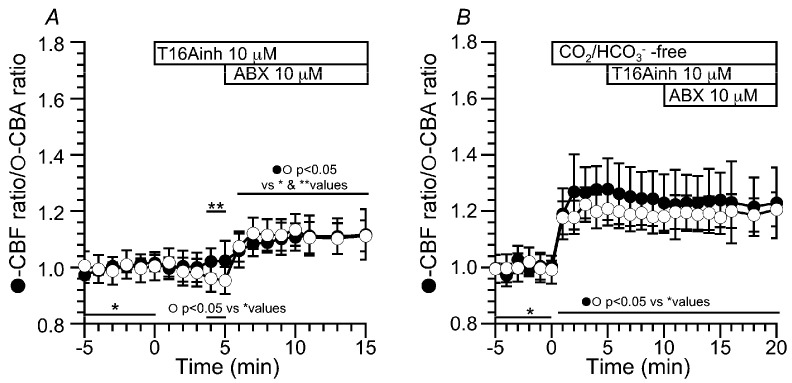
Effects of T16Ainh (an inhibitor of ANO1) on the CBF and CBA stimulated by ABX. (**A**): In the CO_2_/HCO_3_^−^-containing control solution, the addition of T16Ainh (10 µM) decreased the CBA ratio by 5%, but not the CBF ratio. Further ABX stimulation increased the CBF and CBA ratios by 10%. (**B**): The switch to the CO_2_/HCO_3_^−^-free control solution increased the CBF ratio by 30% and the CBA ratio by 20%, and then, the addition of T16Ainh did not change the CBF and CBA ratios. The addition of ABX did not change the CBF and CBA ratios.

**Figure 9 ijms-24-16976-f009:**
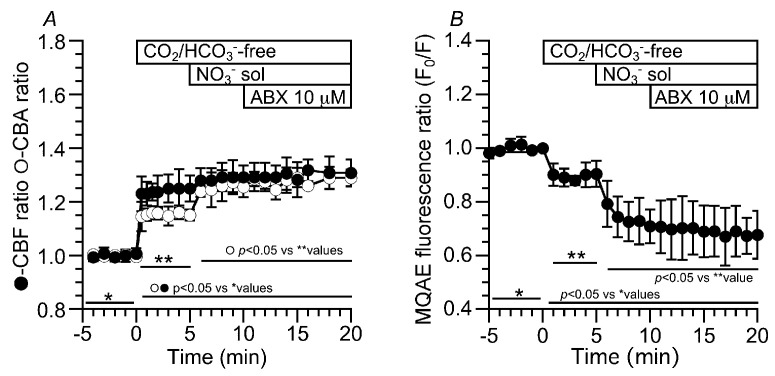
Effects of CO_2_/HCO_3_^−^-free Cl^−^-free NO_3_^−^ solution on the CBF and CBA. (**A**): The switch to the CO_2_/HCO_3_^−^-free solution increased the CBF and CBA ratios and then the switch to the CO_2_/HCO_3_^−^-free Cl^−^-free NO_3_^−^ solution increased the CBA ratio, but not the CBF ratio. Further ABX stimulation did not increase the CBF and CBA ratios. Thus, the increases in CBA and CBF stimulated by ABX were mimicked by the CO_2_/HCO_3_^−^-free Cl^−^-free NO_3_^−^ solution. (**B**): Changes in [Cl^−^]_i_ monitored via the MQAE fluorescence ratio (F_0_/F). The switch to the CO_2_/HCO_3_^−^-free solution decreased F_0_/F and then the second switch to the CO_2_/HCO_3_^−^-free Cl^−^-free NO_3_^−^ solution decreased F_0_/F. Further ABX stimulation did not change F_0_/F.

**Figure 10 ijms-24-16976-f010:**
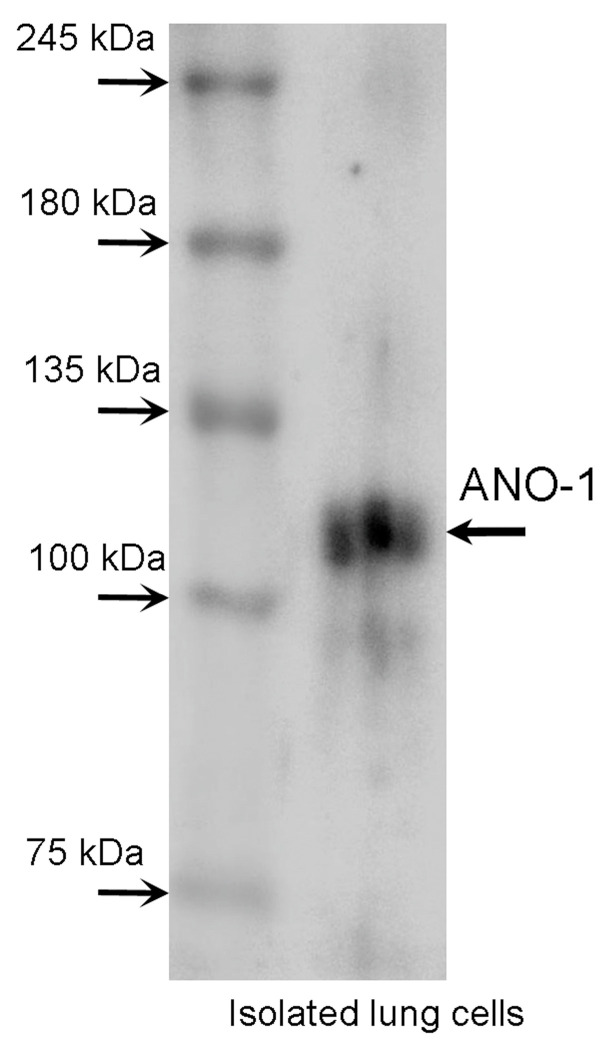
Western blotting for ANO1 in isolated lung cells. A single band for ANO1 was detected at 110 kDa in isolated lung cells.

**Figure 11 ijms-24-16976-f011:**
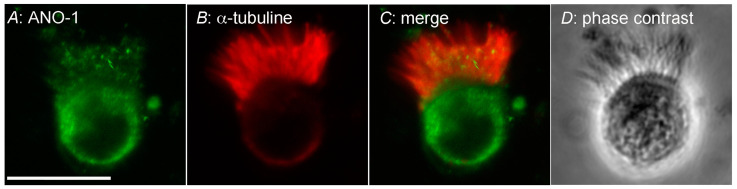
Immunofluorescence examination of ANO1. (**A**): ANO1. (**B**): α-tubuline. (**C**): Merged image. (**D**): Phase contrast image. Cilia were positively stained for ANO1.

**Figure 12 ijms-24-16976-f012:**
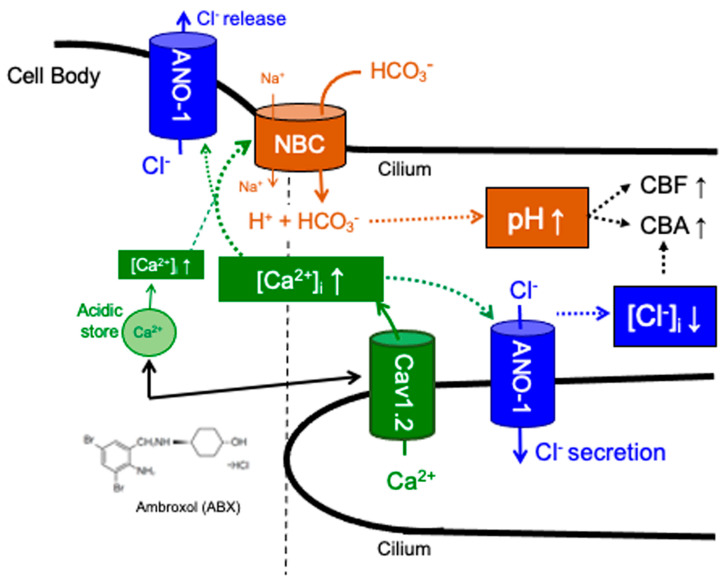
Schematic diagram of the ABX-stimulated c-LAECs. ABX stimulates the Ca^2+^ entry via Ca_V_1.2 and increases [Ca^2+^]_i_. The [Ca^2+^]_i_ increase stimulates the NBC to accelerate HCO_3_^−^ entry. The HCO_3_^−^ entering via NBC binds H^+^ to increase pH_i_. The Ca^2+^ entering via Ca_V_1.2 directly stimulates ANO1 in cilia to activate Cl^−^ secretion, which decreases [Cl^−^]_i_. The pH_i_ elevation enhances the CBF and CBA, and the [Cl^−^]_i_ decrease enhances the CBA. A small and transient Ca^2+^ release from the acidic stores increases [Ca^2+^]_I_ and activates NBC in ABX-stimulated c-LAECs.

## Data Availability

The data that support the findings of this study are in the paper itself, and no shared data are used in the paper.

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
