# Peer review of "Ambroxol-Enhanced Frequency and Amplitude of Beating Cilia Controlled by a Voltage-Gated Ca2+ Channel, Cav1.2, via pHi Increase and [Cl]i Decrease in the Lung Airway Epithelial Cells of Mice"

_ijms, 2023, doi:10.3390/ijms242316976_

Round 1

Reviewer 1 Report

Comments and Suggestions for Authors

The authors examined the mechanisms by which Cav1.2 activation by ambroxol (ABX) increases CBF and CBA in mouse c-LAECs, and found that ABX-induced Cav1.2 activation stimulated NBC followed by pHi increase, which enhanced CBF and CBA, and activated ANO1 followed by [Cl-]i decrease, which enhanced CBA. This is well organized study and clearly showed the data. Some issues should be clarified to improve the manuscripts.

1.    The last sentence in Abstract repeats the similar description of the former sentence. Better to remove the last sentence and add the conclusions of this study.

2.    ABX also inhibits voltage-gated Na channel (Nav), although at higher concentrations than dose inhibiting Cav1.2 as the authors recently stated {ref. 6}. However, since there is no data showing the effects of [Na+]i changes on CBF and CBA (or on pHi or [Cl-]i), please add discussion for non-specific effects of ABX.

3.    Fig. 11 showed the basolateral expression of ANO1 much higher than the ciliary expression of ANO1. Therefore, the involvement of ANO1 cannot be limited to the ciliary ANO1. How about the expression of ANO1 in the tissue section? Is it possible that isolated c-LAECs lost the polarity of ANO1 expression? Please add discussion and/or change the description (L439 ‘The coupling of ANO1 and Cav1.2’ and Fig. 12 schema).

4.    Fig. 2 legend (L162-165); Since no data for Na+/K+ ATPase activity or hyperpolarization in Fig. 2, the authors should not contain the speculation in figure legend. Please remove this sentence.

5.    Some typos are present throughout the manuscript. L29 pHi; L45 acivating; L75 Cav1.2 --- Cav1.2 activation; L327 CaV1.2; L386 Ca2+ relase; Please correct.

Comments on the Quality of English Language

The manuscript was well written. May reduce 'been shown to'.

Reviewer 2 Report

Comments and Suggestions for Authors

The authors aim to elucidate the mechanisms by which Ambroxol (ABX) enhances ciliary beat frequency (CBF) and amplitude (CBA) in mouse lung airway epithelial cells. They hypothesize that ABX activates the Ca2+-activated Cl- channel ANO1, leading to an increased intracellular calcium ([Ca2+]i) and subsequent changes in CBF and CBA. Through a series of experiments involving various inhibitors and conditions, the authors conclude that ABX-mediated activation of ANO1 and subsequent intracellular events are responsible for the observed increases in ciliary motility.

Summary of Major Concerns:

  1. Calcium-Independent Effects of ABX: Figure 2C indicates that ABX's effect on [Ca2+]i is nullified in CO2/HCO3--free conditions. However, Figures 5C and 5D demonstrate that ABX still induces cell shrinkage and a Cl- decrease without CO2 and HCO3-, suggesting a calcium-independent mechanism for ABX's action. This observation challenges the proposed model where an increase in intracellular calcium is essential for ABX's effect and warrants further investigation or clarification.
  2. Direct Regulation by Intracellular Calcium: The direct regulatory effect of intracellular calcium on the outer dynein arm, which affects CBF, has been established in the literature (Inaba K., Cilia 2015). The current manuscript does not seem to consider this direct regulation mechanism, which could be a significant oversight given the manuscript's focus on calcium signaling pathways.
  3. ANO1 Involvement in CBF/CBA Regulation: Data from Figure 8A suggests that ABX can increase CBF and CBA in the presence of an ANO1 inhibitor, which implies a mechanism of action via pHi modulation rather than through ANO1. Additionally, Figure 8B shows that CO2/HCO3--free conditions enhance ciliary motility independently of ANO1 inhibition and ABX, calling into question the necessity of ANO1 involvement in the regulation of CBF and CBA by ABX.

Given these concerns, I recommend that the authors address the following points to strengthen the manuscript:

  • Clarify the role of intracellular calcium signaling in ABX's mechanism of action, particularly in light of the calcium-independent effects observed in CO2/HCO3--free conditions.
  • Discuss the potential direct effects of intracellular calcium on ciliary motility, independent of ANO1 activation and pHi elevation, in the context of the existing literature.
  • Reevaluate the data supporting ANO1's involvement in ABX-induced changes in ciliary motility, especially considering the evidence suggesting alternative pathways for ABX's action.

I believe that addressing these issues will significantly enhance the clarity and impact of the findings.

Comments on the Quality of English Language

Minor points: The Results section could benefit from reorganization to improve clarity. Presenting experiments in a logical sequence, grouping related experiments, using descriptive subheadings, and summarizing key findings would greatly aid reader comprehension. Additionally, clear linking statements that connect the outcomes of successive experiments would further underscore the support for the paper's overarching hypothesis.

Typos and Grammatical Corrections:

  • Line 45: "acivating" should be "activating".
  • Line 151: "decreaseing" should be "decreasing".
  • Line 171: "(n=4, Fig. 3B))" should be "(n=4, Fig. 3B)".
  • Line 196: "vido" should be "video".
  • Line 265: "anf" should be "and".
